# A new Gaussian curvature of the image surface based variational model for haze or fog removal

**Muhammad Arif**[1], **Noor Badshah**[1], **Tufail Ahmad Khan**[1], **Asmat Ullah**[1]*, **Hena Rabbani**[1‡], **Hadia Atta**[2‡], **Nasra Begum**[3‡]

**1** Department of Basic Sciences, University of Engineering and Technology, Peshawar, Pakistan,
**2** Department of Mathematics, Islamia College Peshawar, Peshawar, Pakistan, **3** Department of Mathematics, Shaheed Benazir Bhutto Women University, Peshawar, Pakistan

☯ These authors contributed equally to this work.
‡ HR, HA and NB also contributed equally to this work.
* asmatullah75@gmail.com

**Data Availability Statement:** All relevant data is available within the paper.

**Funding:** All authors themselves will contribute for funding of the manuscript publication fee.

## Abstract

Outdoor images are usually affected by haze which limits the visibility and reduces the contrast of the images. Removal of haze from real-world images is always a challenging task. Recently, many mathematical models have been proposed for the effective removal of haze from real-world images. However, these models may produce staircase effects or lower the image contrast or smooth the edges of the object. In this paper, we propose a model based on Gaussian curvature for the de-hazing of images. The atmospheric veil estimate is included based on dark channel prior (DCP), which can significantly reduce the artifacts on the edge of the image and increase the accuracy. The transmission map then changes to a high-quality map to reduce haze or fog from gray and color images. DCP combined with Gaussian curvature is done for the first time for image de-hazing/de-fogging. The augmented Lagrangian method is used to find the minimizer of the proposed functional, which will be a system of partial differential equations. To get fast convergence, fast Fourier transforms (FFT) is used to solve the system of PDEs. The performance of the proposed model is compared with other state-of-the-art models qualitatively and quantitatively. The proposed model is tested on various real and synthetic images which show better efficiency in staircase effects reduction, haze/fog removal, image contrast, corners, and sharp edges conservation respectively.

## 1 Introduction

The atmospheric light is dispersed in different angles due to the presence of atmospheric particles (e.g. fog, haze, smog, smoke, and mist) in the air. The incoming light blended to the layer of ambient light (air-light), which reflects atmospheric particles in the line of sight, depending on the turbidity and distance from the scene to the observer (visual range), causing low contrast, reduced visibility, distortion of colour, deterioration of essential elements of the

**Competing interests:** The authors have declared that no competing interests exist.

photography scene and makes lots of trouble in the recognition and detection of a target in video surveillance system by blocking the direct scene transmission. Therefore, it is primal imperative to improve visibility and restore the essential features of the scene with a simple and effectual image restoration algorithm.

The removal of haze or fog is considered to be an important procedure since haze or fog-free images are visually pleasing and can improve performance of computer vision tasks like object detection, classification, visual navigation, etc. According to Koschmieder [1], a degraded scene formed as displayed in Fig 1 can be formulated mathematically as follows

$$I(x) = T(x)J(x) + I_\infty(-T(x) + 1), \tag{1}$$

where the observed hazy or foggy image intensity is $I(x)$, the intensity of the scene is $J(x)$, the intensity of the atmospheric light is $I_\infty$ and $T(x) = e^{-\beta d(x)}$ is the transmission map (scene reflected light captured by the signal receiver depending upon the amount of haze) corresponds to values between 0 (no visibility) and 1 (clear visibility) with degradation coefficient $\beta$ and distance $d(x)$ from scene point to the observer. The first term $T(x)J(x)$ in the R.H.S of Eq (1) is the direct attenuation, describes the radiance of the scene while the second term $I_\infty(-T(x) + 1)$ is the air-light (result of the scattered atmospheric light due to atmospheric particles), twists the radiance of the scene. The key task to get **J** from an observed hazy frame I is the T estimate and $I_\infty$ atmospheric light. Many researchers have made significant progress in the estimation of transmission T using a single image [2–4] and took the average value from I or I as the atmospheric light $I_\infty$ as its brightest pixel.

It should be noted that inaccurate assumption or estimation of atmospheric light $I_\infty$ leads towards imprecise results in the recuperation of **J** haze free image. Without the use of rational processes, several methods assume the pixels with dense fog to be pure white and take the transmissions in a direct relation with hazy colors [5–7]. Due to providing unsatisfactory transmission map they proceed towards Laplacian matting algorithm or guided filter algorithm for refinement. The inappropriate assumption of linearity between transmission map and haze colors applying by these two algorithms reflects every variance of hazy colors on transmission map without difference. Taking advantages of the physical properties of air-light map or atmospheric veil, Cho et al. [9] used a variational approach for the estimation of air-light map in order to restore a fog-free image. This approach can adequately remove the haze and satisfy the edge preserving property, but the restored images have very low contrast. For this purpose, a variety of histogram equalization methods were applied to make better the contrast of the restored images. During image de-hazing, it should be take into account that the accurate estimation of depth or medium transmission information leads towards finest de-hazing results. Most de-hazing procedures are based on the pre-estimated depth or medium

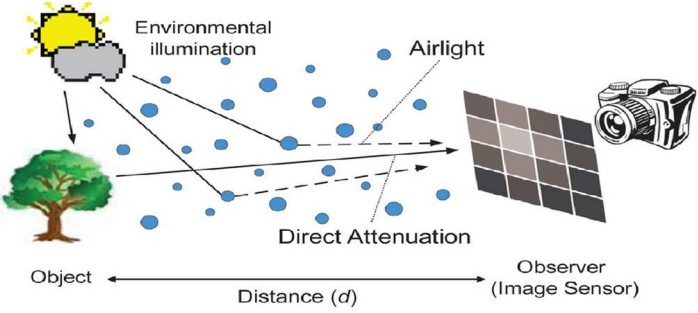

**Fig 1. Image formation model in turbid medium [8].**

transmission information. Fang et al. [10] estimated the depth map and de-haze image using a total variation (TV) regularizer and [11, 12] used the TV regularizer to medium transmission refining. TV regularizer has the advantage of preserving edges of the image but produces blocky effects due to piecewise constant solutions where the region is homogeneous. Total generalised variation (TGV) second order regularizer was implemented to overcome the blocking artifacts which has the ability of reducing stair-casing effects along with edge preservation [13, 14].

The de-hazing methods are mainly classified in methods based on enhancement and physics. Methods based on enhancement do not take the mechanism of image degradation into account. They simply intend to make better the degraded image contrasts and ignore the colour restoration. The simple, fast and most widely used enhancement-based methods are typical retinex [15] and histogram-based method [16]. These methods may not properly work to remove visual defects of the image having inhomogeneous fog. The restored image can be restored overly and contains halo objects.

Physics-based methods design a physical model based on a hazy image's degradation phenomenon. Then, using that physical model, restore the haze-free image. In contrast of methods based on enhancement, physics-based methods yield natural, finest and accurate de-hazing results that thrust someone into attention as explained in many scientific theories and experimental results. Therefore, the main subject of this paper is the dehazing processes based on the physical model. In physics-based de-hazing methods, the dark channel prior [17] is most widely used and assuming that there are some dark pixels have a very low intensity of approximately zero in at least one of the RGB color channels in the non-sky patches of a haze free scene. Since the DCP is based on the non-sky regions, therefore it gives very poor results for images with sky regions. He et al. [17] used the soft matting algorithm to annihilate halo artifacts generated by the kernel operation and refined the transmission map. They improved the de-hazing results very well but the soft matting algorithm is very costly and time consuming. He et al. [18] also employed guided filtering in place of soft matting algorithm for transmission map optimization. The guided filtering gives similar results as that of soft matting algorithm but the de-hazing speed is very fast. The de-hazing performance of DCP is further improved by Xu et al. [19] and Li et al. [20]. Color lines in RGB space can also better approximate pixels as shown by [21–23]. Tarel et al. [24] assumed the atmospheric veil to be changed easily in a local medium and applied the median filter to improve the proposed de-hazing results. The median filter can't conserve the object edges well, therefore, in case of discontinuous scene depth it may fail. Meng et al. [25] used a boundary constraint and $L_1$-norm-based regularization to estimate and refine the transmission map. They became successful in reducing halo artifacts but the de-hazed results have some color distortion. Nishino et al. [26] and Wang et al. [27] used Bayesian probabilistic method and Bayesian theory based on the Markov random field to calculate the scene albedo and scene depth to remove haze. By reducing the kernel operation using Bayesian-based algorithms, halo artifacts can be reduced and the problem of color distortion occurs as in Meng et al. [25]. Ancuti and Ancuti [28] employed a Laplacian pyramid and a Gaussian pyramid and proposed a multi-scale fusion-based haze removal algorithm. The algorithm have fast processing speed but the resulting image have high saturation and color distortion due to not focusing on the distribution of haze amount relative to the scene depth. Lee et al. [31] stated that only Gaussian curvature can maintain important structures (such as corners(edges) and wrinkles). The work was supported by Elsey and Esedoglu [29] and proved that the Gaussian curvature of the surface is a good preserver for important features of image as compared to mean curvature.

Many haze removal techniques have been proposed so far. The problems of halo artifacts, stair-cases like artifacts, and color distortion that need to be solved during the image dehazing/

defogging process are still unsolved. In this regard, our contributions to tackling such issues can be outlined as follows:

- A new variation model based on the Gaussian curvature (GC) of the image surface of a given scene is proposed, to estimate an air-light map which preserves the important image features such as edges and wrinkles while handling the problems of halo artifacts, stair case effects and color distortions simultaneously quite well.

- We also take some prior assumptions called the dark channel prior (DCP) to approximate the transmission map. The transmission map is then changed into a depth map.

- Dark channel prior combined with Gaussian curvature is employed for the first time for image de-hazing/de-fogging.

- We implement augmented Lagrangian method (ALM) for the Gaussian curvature (GC) regularization based de-hazing model and design a special minimization procedure to minimize the augmented Lagrangian functional.

- For fast convergence, fast Fourier transform is implemented to compute the system of linear partial differential equations arisen from the minimization of the augmented Lagrangian functional.

This paper is summarized in the following. In section 2, a review of some well-known image de-hazing and defogging related methods is given. In section 3, our novel model for addressing the problem is proposed. The augmented Lagrangian method is used in Section 4 for the resolution of the Euler Lagrange PDEs system. Section 5 provides some image restoration and analysis results for outdoor and indoor scenes to demonstrate the performance of the proposed model. Lastly, we make some concluding remarks, applications and future directions in section 6. Next, we discuss some well known related image de-hazing and defogging methods as under.

## 2 Preliminaries

In this section, the following two well-known recent image de-hazing or defogging methods are reviewed. The restoration results of these have been compared with the proposed model discussed in subsection-5.4.

### 2.1 Edge preserving regularization based single image de-fogging model (M1)

In order to recover a fog-free scene, Cho et al. [9] (M1), based on the physical properties of air-light $A$, proposed a model for estimating the air-light map $A$. The energy functional of the model is

$$E(A) \quad = \int_{\Omega} H(-(A - W))dxdy + \lambda^2 \int_{\Omega} (\psi \parallel \nabla A \parallel)dxdy \tag{2}$$

where data fidelity term is the first term and a regularization term is the second term. $W$ = min $(I)$ is the image of minimal component of image $I$, $H(\cdot)$ is the Heaviside function, $\Omega$ is bounded open subset of $R^2$, $||\nabla A||$ is the $A$ gradient modulus, and $\lambda$ is the regularization parameters that balance the influence between two terms of Eq (2). The function $\psi$ is presumed to be an even function of class $C^2(R)$ as the edge preserving regularization function.

The following slightly regularized versions of the function $H$ and $\delta$ (one-dimensional Dirac measure), denoted here by $H_\epsilon$ and $\delta_\epsilon = H'_\epsilon$, are considered.

$$H_\epsilon(z) = \frac{\pi + 2\arctan\left(\frac{z}{\epsilon}\right)}{2\pi}, \quad \delta_\epsilon(z) = \frac{\epsilon}{\pi(\epsilon^2 + z^2)}$$

As $\epsilon \to 0$, they converge respectively to $H$ and $\delta$. The method improves the quality of the defogged image significantly and restores a fogged image in better colour. The algorithm is good even in case of heavy fog. However, without using histogram for the post-processing of the restored image, the image is not as bright as the atmospheric light, since it does not normally have the same brightness(taken by top 1% brightness pixel values) [9].

## 2.2 Dark channel prior based single image haze removal model (M2)

Dark Channel Prior (DCP) [4] (M2) is based on the assumption that in at least one of the non-sky space patches of a hazelnut free image there are some very low intensity dark pixels equal to zero. The $J$ image can be formulated as following with the dark $J^{dark}$ channel:

$$J^{dark}(x) = \min_{c \in \{R,G,B\}} \left( \min_{y \in \Omega^*(x)} J^c(y) \right) \approx 0, \tag{3}$$

where $J^c$ is a single band (color) image of $J(x)$, $c \in \{R, G, B\}$ is a colour channel, and $\Omega^*(x)$ is a local patch centered at pixel $x$ and the DCP represents the approximate value for every pixel of the dark channel.

**Estimating the transmission map.**    Assuming $I_\infty$ to be known. According to the DCP approximation, the patch's transmission $\widehat{T}(x)$ can be represented as:

$$\widehat{T}(x) = -\min_{c \in \{R,G,B\}} \left( \min_{y \in \Omega^*(x)} \frac{I^c(y)}{I_\infty^c} \right) + 1. \tag{4}$$

Since, the colour of the sky nearly approach to $I_\infty$ in hazy scenes and on removing fog or haze thoroughly may result un-natural scenes [4, 17] and depth information may also be lost.

$$\min_c \left( \min_{y \in \Omega^*(x)} \frac{I^c(y)}{I_\infty^c} \right) \approx 1 \text{ and } \widehat{T}(x) \approx 0. \tag{5}$$

A constant $\omega \in (0, 1]$ is multiplied to prevent the depth information of the natural scene and is generally fixed as 0.95. Considering that the transmission is a constant in $\Omega^*(x)$, and performing the important operation in the local patch and three color channels on the haze scene $I$. The estimated value of $T(x)$ defined by $\widehat{T}(x)$ can be computed as follows

$$\hat{T}(x) = -\omega \, \min_c \left( \min_{y \in \Omega^*(x)} \frac{I^c(y)}{I_\infty^c} \right) + 1. \tag{6}$$

This process of estimating the transmission map is sensibly well. As in local patches, the transmission is taken constant which generates some blocky artifacts because the fact is that in a patch, the transmission is not always constant. Furthermore, the transmission map is refined through soft matting process.

**Soft matting.**    He et al. [17] employed a soft matting algorithm [30] for refining the transmission map $T(x)$ and minimized the following cost function:

$$E(\mathbf{T}) = \mathbf{T}^t L \mathbf{T} + \lambda (\mathbf{T} - \hat{\mathbf{T}})^t (\mathbf{T} - \hat{\mathbf{T}}), \tag{7}$$

where $\mathbf{T}$ and $\hat{\mathbf{T}}$ are the vectorial forms of $T(x)$ and $\hat{T}(x)$, $\lambda$ is a regularization parameter and $L$ is the Matting Laplacian matrix whose (i,j) element is computed as:

$$\sum_{k|(i,j)\in w_k} (\delta_{i,j} - \frac{1}{|w_k|}(1 + (I_i - \mu_k)^t(\Sigma_k + \frac{\varepsilon}{|w_k|}U_3)^{-1}(I_j - \mu_k))),\tag{8}$$

where $\delta_{i,j}$ is the Kronecker delta, $\mu_k$ and $\Sigma_k$ are the average and co-variance matrix of the colours in windows $w_k$, $I_i$ and $I_j$ are the colours of the scene $I$ at pixels $i$ and $j$, $|w_k|$ is the number of pixels in $w_k$, $\varepsilon$ is a regularization parameter and $U_3$ ia a $3 \times 3$ identity matrix.

The optimal value of $\mathbf{T}$ is the following sparse linear system solution:

$$(L + \lambda U)\mathbf{T} = \lambda \hat{\mathbf{T}},\tag{9}$$

where $U$ is an identity matrix equal with $L$.

**Recovering the scene radiance.**   Once the transmission map $T(x)$ and atmospheric light $I_\infty$ is known, the scene radiance $J(x)$ is computed from Eq (1) by:

$$J(x) = I_\infty + \frac{-I_\infty + I(x)}{max(T(x), T_0)},\tag{10}$$

where $T_0$ represents a lower bound of $T(x)$ and its optimal value is nearly 0.1. The scene radiance has low brightness as compared to atmospheric light, the restored images seem to be faint and the exposure of $J(x)$ is increased for displaying.

**Approximating the atmospheric light.**   To improve the atmospheric light estimation using the DCP (more faster than the "brightest pixel" method), the pixels with (highest) intensity in the top 0.1% most haze opaque brightest pixels in the hazy scene $I$ is selected as the atmospheric light $I_\infty$. Although single image haze removal can easily enhance the appearance of images applying DCP on the degradation model but it may not work on images having scene object similar to the atmospheric light in an inherent manner and no shadow is diffused on them. The DCP underrate the transmission for these objects and some halo artifacts are also found in the resulting images. To overcome the above-mentioned drawbacks of M1 and M2 methods, a new variational model for haze or fog removal is proposed as follows:

## 3 The proposed de-hazing model (M3)

In this section, we propose a new model based on dark channel prior (a kind of statistics of the haze/fog-free outdoor scenes) and using Gaussian curvature (GC) as a regularizer. Dark channel prior depends on key perception-most local patches in haze/fog-free (outdoor) images contain a few pixels which have extremely low intensities in at least one color channel. Utilizing this prior to the haze/fog imaging model, we can straightforwardly approximate the thickness of the haze and recuperate a top-quality haze/fog-free image. This model uses advantages of both DCP and GC, which leads to good restoration results, because dark channel prior performs well for estimating the thickness of the haze/fog and recovering good quality scene, while Gaussian curvature performs well for eliminating staircase effect, preserving textures and sharp edges. The resulting model has the advantage of better preserving image regions containing textures and fine details, leading to a more natural scene while reducing the staircase effect in smooth regions.

Hence motivated by the advantages of Gaussian curvature compared to the mean curvature and total variation in 2D image de-noising pointed out by Elsey and Esedoglu [29] and Lee and Seo [31] in geometry processing and dark channel prior, here we design a Gaussian curvature of the scene surface regularization and DCP based model for single image de-hazing.

Hence employing this model, reasonable improvement in image quality is obtained. The Gaussian curvature of a 3D surface $S$ can be clearly expressed by the zero-level set function $\Phi$ and can be defined by

$$k_{GC} = \frac{\nabla\Phi H^*(\Phi)\nabla\Phi^t}{|\nabla\Phi|^4},$$

(11)

where $\nabla\Phi = (\Phi_x, \Phi_y, \Phi_z)^t$ is the gradient vector, $|\nabla\Phi| = \sqrt{\Phi_x^2 + \Phi_y^2 + \Phi_z^2}$ is the norm

of $\nabla\Phi$, $H(\Phi) = \begin{pmatrix} \Phi_{xx} & \Phi_{xy} & \Phi_{xz} \\ \Phi_{yx} & \Phi_{yy} & \Phi_{yz} \\ \Phi_{zx} & \Phi_{zy} & \Phi_{zz} \end{pmatrix}$ and $H^*(\Phi) =$

$\begin{pmatrix} \Phi_{yy}\Phi_{zz} - \Phi_{yz}\Phi_{zy} & \Phi_{yz}\Phi_{zx} - \Phi_{yx}\Phi_{zz} & \Phi_{yx}\Phi_{zy} - \Phi_{yy}\Phi_{zx} \\ \Phi_{xz}\Phi_{zy} - \Phi_{xy}\Phi_{zz} & \Phi_{xx}\Phi_{zz} - \Phi_{xz}\Phi_{zx} & \Phi_{xy}\Phi_{zx} - \Phi_{xx}\Phi_{zy} \\ \Phi_{xy}\Phi_{yz} - \Phi_{xz}\Phi_{yy} & \Phi_{xz}\Phi_{yx} - \Phi_{xx}\Phi_{yz} & \Phi_{xx}\Phi_{yy} - \Phi_{xy}\Phi_{yx} \end{pmatrix}^t$ are the Hessian matrix and its

adjoint, respectively.

Let us consider the surface $S$ to be the graph of a hazy image $I$ and the unexplored atmospheric region between the scene and sensor be the channel of air-light mapping $(x, y)$: $\rightarrow (x, y, A(x, y))$. Then the relation $\Phi = -z + A(x, y)$ can be used to find a formula for $k_{GC}$. The new coordinate system implicates $\nabla\Phi = (A_x, A_y, -1)^t$,

$$H(\Phi) = \begin{pmatrix} A_{xx} & A_{xy} & 0 \\ A_{yx} & A_{yy} & 0 \\ 0 & 0 & 0 \end{pmatrix},$$

$$H^*(\Phi) = \begin{pmatrix} 0 & 0 & 0 \\ 0 & 0 & 0 \\ 0 & 0 & A_{xx}A_{yy} - A_{xy}A_{yx} \end{pmatrix}.$$

The GC of a 2D image surface may be formulated as

$$k_{GC} = \frac{A_{xx}A_{yy} - A_{xy}A_{yx}}{(A_x^2 + A_y^2 + 1)^2}.$$

(12)

Now, it is possible to explicate a new image de-hazing model using the following regularizer

$$R^{GC}(A) = \int_\Omega \left| \frac{A_{xx}A_{yy} - A_{xy}A_{yx}}{(A_x^2 + A_y^2 + 1)^2} \right| dxdy$$

$$= \int_\Omega \left| \frac{\det(H(A))}{(|\nabla A|^2 + 1)^2} \right| dxdy.$$

(13)

Therefore, our new designed Gaussian curvature based image de-hazing model is

$$\min_A F_f^r(A) = \int_\Omega H(W - A)dxdy + \lambda^2 R^{GC}(A),$$

(14)

where the λ parameter is positive and $R^{GC}(A)$ is the Gaussian curvature regularization term. To minimize the Gaussian curvature term, let us take

$$\delta R^{GC} = \frac{d}{d\epsilon} R^{GC}(A + \epsilon\Phi)|$$

$$= \left[\frac{d}{d\epsilon} \int_\Omega \left|\frac{(A+\epsilon\Phi)_{xx}(A+\epsilon\Phi)_{yy} - (A+\epsilon\Phi)_{xy}(A+\epsilon\Phi)_{yx}}{\left((A+\epsilon\Phi)_x^2 + (A+\epsilon\Phi)_y^2 + 1\right)^2}\right| dxdy\right]_{\epsilon=0}$$

$$= \int_\Omega \frac{A_{xx}A_{yy} - A_{xy}A_{yx}}{|A_{xx}A_{yy} - A_{xy}A_{yx}|} \cdot \frac{(A_{xx}\Phi_{yy} + A_{yy}\Phi_{xx} - A_{xy}\Phi_{yx} - A_{yx}\Phi_{xy})}{(A_x^2 + A_y^2 + 1)^2} dxdy$$

$$- \int_\Omega \frac{4|A_{xx}A_{yy} - A_{xy}A_{yx}|(A_x\Phi_x + A_y\Phi_y)}{(A_x^2 + A_y^2 + 1)^3} dxdy$$

Here, we apply some new notations ($\Gamma = 1 + A_x^2 + A_y^2$, $S = sign(A_{xx}A_{yy} - A_{xy}A_{yx})$ where sign (○) is the sign function and $(v_1, v_2) = v$ the unit normal vector) to reduce the complexity in writing the equations and use the divergence theorem when required.

$$\delta R^{GC} = -\int_\Omega \Phi\left(\frac{SA_{xy}}{\Gamma^2}\right)_{xy} - \int_{\partial\Omega} \Phi_y \frac{SA_{xy}}{\Gamma^2} v_1 d\Gamma + \int_{\partial\Omega} \Phi\left(\frac{SA_{xy}}{\Gamma^2}\right)_x v_2 d\Gamma$$

$$- \int_\Omega \Phi\left(\frac{SA_{yx}}{\Gamma^2}\right)_{yx} - \int_{\partial\Omega} \Phi_x \frac{SA_{yx}}{\Gamma^2} v_2 d\Gamma + \int_{\partial\Omega} \Phi\left(\frac{SA_{yx}}{\Gamma^2}\right)_y v_1 d\Gamma$$

$$- \int_\Omega \Phi\left(\frac{SA_{xx}}{\Gamma^2}\right)_{yy} + \int_\Omega \Phi_y \frac{SA_{xx}}{\Gamma^2} v_2 d\Gamma - \int_\Omega \Phi\left(\frac{SA_{xx}}{\Gamma^2}\right)_y v_2 d\Gamma$$

$$- \int_\Omega \Phi\left(\frac{SA_{yy}}{\Gamma^2}\right)_{xx} - \int_{\partial\Omega} \Phi_x \frac{SA_{yy}}{\Gamma^2} v_1 d\Gamma - \int_{\partial\Omega} \Phi\left(\frac{SA_{yy}}{\Gamma^2}\right)_x v_1 d\Gamma$$

$$+ \int_\Omega \Phi\left(\frac{4|A_{xx}A_{yy} - A_{xy}A_{yx}|A_x}{\Gamma^3}\right)_x - \int_{\partial\Omega} \left(\frac{4|A_{xy}^2 - A_{xx}A_{yy}|A_x}{\Gamma^3}\right) v_1 d\Gamma$$

$$+ \int_\Omega \Phi\left(\frac{4|A_{xx}A_{yy} - A_{xy}A_{yx}|A_y}{\Gamma^3}\right)_y - \int_{\partial\Omega} \left(\frac{4|A_{xx}A_{yy} - A_{xy}A_{yx}|A_y}{\Gamma^3}\right) v_2 d\Gamma.$$

In a state of proper arrangement, we drop the boundary terms.

$$(-A_{xy}, \ A_{xx}).v = 0, \quad (A_{yy}, -A_{yx}).v = 0,$$

$$\left(\left(\frac{SA_{yx}}{\Gamma^2}\right)_y, \ -\left(\frac{SA_{xx}}{\Gamma^2}\right)_y\right).v = 0, \quad \left(-\left(\frac{SA_{yy}}{\Gamma^2}\right)_x, \ \left(\frac{SA_{xy}}{\Gamma^2}\right)_x\right).v = 0.$$

Eventually, we define

$$\beta_1 = \left(\left(\frac{SA_{yy}}{\Gamma^2}\right)_x, -\left(\frac{SA_{xy}}{\Gamma^2}\right)_x\right), \tag{15}$$

$$\beta_2 = \left(-\left(\frac{SA_{yx}}{\Gamma^2}\right)_y, \ \left(\frac{SA_{xx}}{\Gamma^2}\right)_y\right), \tag{16}$$

and the Euler Lagrange equation with the above defined boundary conditions for the GC

based image de-hazing model can be written as

$$\lambda^2 \nabla \cdot \left( \frac{4|A_{xx}A_{yy} - A_{xy}A_{yx}|}{\Gamma^3} \nabla A \right) + \nabla \cdot \beta_1 + \nabla \cdot \beta_2 - \delta(W(x,y) - A(x,y)) = 0. \qquad (17)$$

To solve the non-linear PDE (17), an augmented Lagrangian method is employed as discussed in the next section.

## 4 Numerical implementation

Augmented Lagrangian method (ALM) plays an important role in solving constraint minimization problems and is used in several image restoration problems such as [32–38]. ALM convert the constraint minimization problem to unconstraint problem by incorporating the constraints in the energy functional, on account of which some additional terms arise known as Lagrange multiplier terms. In ALM, the problem is classified into different sub-problems using alternating minimization procedure that can be solved easily.

**Notation.** Representing a gray-scale image by an $N^2$ matrix and Euclidean space $\mathbb{R}^{N \times N}$ by $V$. The gradient operator (discrete) is a mapping $\nabla \colon V \to Q = V^2$. For $A \in V$, $\nabla A$ is computed by:

$$(\nabla A)_{i,j} = ((\mathring{D}_x^+ A)_{i,j}, (\mathring{D}_y^+ A)_{i,j}),$$

with

$$(\mathring{D}_x^+ A)_{i,j} = \begin{cases} -A_{i,j} + A_{i+1,j}, & N-1 \geq j \geq 1 \\ A_{i,1} - A_{i,N}, & N = j \end{cases}$$

$$(\mathring{D}_y^+ A)_{i,j} = \begin{cases} -A_{i,j} + A_{i,j+1}, & N-1 \geq j \geq 1 \\ A_{1,j} - A_{N,j}, & j = N, \end{cases}$$

where $i, j = 1, \ldots, N$. $\mathring{D}_x^+$ and $\mathring{D}_y^+$ indicate forward difference operators with boundary condition (periodic)($A$ is periodically extended).We represent the usual inner product of $V$ and $Q$ as $\langle \cdot, \cdot \rangle_V$ and $\langle \cdot, \cdot \rangle_Q$ and the Euclidean norm of $V$ and $Q$ by $\|\cdot\|_V$ and $\|\cdot\|_Q$ that can be defined as follows:

$$(p, q)_Q = (p^1, q^1)_V + (p^2, q^2)_V,$$

$$\| p \|_Q = \sqrt{(p, p)_Q},$$

for $p = (p^1, p^2) \in Q$ and $q = (q^1, q^2) \in Q$. Moreover, at each pixel $(i, j)$,

$$|p_{i,j}| = |(p_{i,j}^1, p_{i,j}^2| = \sqrt{(p_{i,j}^1)^2 + (p_{i,j}^2)^2},$$

is the usual Euclidean norm in $\mathbb{R}^2$. For the sake of convenience, we neglect the subscripts $V$ and $Q$ and simply use $(\cdot, \cdot)$ and $\|\cdot\|$ to represent the usual inner product and $L^2$ norm. For solving the GC de-hazing model (14) with ALM, we are introducing a new dual variable $q$ where $q$

$= \nabla A$, and come into possession of the following refined constrained optimization problem:

$$\min_{A,q} F_f^r(A, q) = \mathbb{D}(W, A) + \lambda^2 R(q) \tag{18}$$

$$s.t \quad q = \nabla A.$$

The constraint minimization problem $F_f^r(A, q)$ is further reformulated to get the augmented Lagrangian functional as follows:

$$\mathbb{E}_{GC}^{rf}(A, q; \mu) = H(W - A) + \lambda^2 R(q) + \langle \mu, q - \nabla A \rangle + \frac{r}{2} \| q - \nabla A \|_2^2, \tag{19}$$

where $\mu$ is the multiplier of Lagrange and $r$ is a constant(positive). The alternating minimization procedure classify the functional into two sub-problems in order to find the optimal values of $A$, $q$ and $\mu$.

## 4.1 Sub-problem for q

For a given $A$ and $\mu$, the functional (objective) is defined by

$$\min_q \lambda^2 R(q) + \langle \mu_k, q \rangle + \frac{r}{2} \| q - \nabla A \|^2. \tag{20}$$

The Euler Lagrange equations for the functional Eq (20), with $\Gamma = q_1^2 + q_2^2 + 1$ is

$$-\left( \left( \frac{(q_2)_y}{\Gamma^2} \right)_x + \left( \frac{(q_2)_x}{\Gamma^2} \right)_y \right) - \frac{4SDq_1}{\Gamma^3} + \mu_1 + r(q_1 - A_x) = 0, \tag{21}$$

$$-\left( \left( \frac{(q_1)_y}{\Gamma^2} \right)_x + \left( \frac{(q_1)_x}{\Gamma^2} \right)_y \right) - \frac{4SDq_2}{\Gamma^3} + \mu_2 + r(q_2 - A_y) = 0, \tag{22}$$

where

$$D = \det(\nabla(q)) = -(q_1)_y(q_2)_x + (q_1)_x(q_2)_y,$$

$$S = sign\left( \frac{D}{(1 + \| \nabla A \|^2)^2} \right).$$

Without applying any iterative procedures, it is very easy to solve Eqs (21) and (22) for $q_1$ and $q_2$.

## 4.2 Sub-problem for A

For a $q$ and $\mu$ given, we have

$$\min_A H(W - A) - \langle \mu_k, \nabla A \rangle + \frac{r}{2} \| q - \nabla A \|^2. \tag{23}$$

The optimality condition for Eq (23) is defined by the linear PDE

$$-r \Delta A - \delta(W - A) + \nabla \cdot \mu_k + r \nabla \cdot q = 0, \tag{24}$$

with the periodic boundary conditions. We implement Fourier transform (and FFT implementation) to compute the above linear equation by following [36, 39–41]. The Fourier

transform of Eq (24) is

$$r\mathcal{F}(\Delta)\mathcal{F}(A) = \mathcal{F}(\mathring{D}_x^-)(\mathcal{F}(\mu_1) + r\mathcal{F}(q_1)) + \mathcal{F}(\mathring{D}_y^-)(\mathcal{F}(\mu_2) + r\mathcal{F}(q_2)) - \mathcal{F}(\delta(W - A)), \quad (25)$$

whose solution is

$$A = \mathcal{F}'\left(\frac{\mathcal{F}(\mathring{D}_x^-)(\mathcal{F}(\mu_1) + r\mathcal{F}(q_1)) + \mathcal{F}(\mathring{D}_y^-)(\mathcal{F}(\mu_2) + r\mathcal{F}(q_2)) - \mathcal{F}(\delta(W - A))}{r\mathcal{F}(\Delta)}\right), \quad (26)$$

where $\mathcal{F}$ and $\mathcal{F}'$ are Fast Fourier transform and inverse Fast Fourier transform, $\mu_k = (\mu_1, \mu_2)$ and $q = (q_1, q_2)$; and Fourier transforms of operators such as $\mathring{D}_x^-$, $\mathring{D}_y^-$, $\Delta = \mathring{D}_x^-\mathring{D}_x^+ + \mathring{D}_y^-\mathring{D}_y^+$ are treated as the transforms of their corresponding convolution kernels. The iterative procedure shown in Algorithm 1 is used to solve Eq (19).

Algorithm 1. ALM for the Gaussian Curvature Based Image Restoration Model

```
1. Initialize μ₁ = μ₂ = 0, A⁽⁰⁾ = min_{c∈{R,G,B}}I^c(x), r ∈ (0, 0.999) and ε = β ∈
(0, 1).
2. For K = 0, 1, ..., IMAX
   (a) Step 1: Solve Eqs (21) and (22) for q₁⁽ᵏ⁺¹⁾ and q₂⁽ᵏ⁺¹⁾ with A = A⁽ᵏ⁾
   (b) Step 2: Solve Eq (26) for A⁽ᵏ⁺¹⁾ with (q₁,q₂) = (q₁⁽ᵏ⁺¹⁾,q₂⁽ᵏ⁺¹⁾)
   (c) Step 3: Update Lagrange multipliers automatically at every
iteration.
μ⁽ᵏ⁺¹⁾ = r(q⁽ᵏ⁺¹⁾ − ∇A) + μ⁽ᵏ⁾
3. End Procedure
```

## 5 Experimental results and analysis

Some images are presented in this section to show the improved performance and efficiency of the proposed model in showing the corresponding visible edge maps, transmission maps, air-light maps, and final de-hazing results. In some scenes (outdoor and indoor) contaminated with haze or fog we describe and analyze the simulation results. The de-hazing results are then comparable to current state of the art methods and corresponding visible edge maps. For quantitative analysis, different quality descriptors/indicators (measures) are taken to assess the amount of newly visible edges, contrast, and mean visibility enhancement restoration. The entropy of the de-hazed images is also computed and compared with other recent methods. In MATLAB R2013a all of the experiments listed here and all simulations have been carried out on a Haier Win8.1 PC, Intel Core i3 CPU @ 1.70GHz with 4.00GB RAM. For better results, the values of Lagrange multipliers $\mu_1$ and $\mu_2$ are tuned automatically at every iteration until optimal restoration results are obtained and choose adaptively $r \in (0, 0.999)$ and $\epsilon = \beta \in (0, 1)$ according to the image size and type.

In this paper, four measures (indicators or descriptors) are taken to evaluate quantitatively the de-hazing outputs of the proposed model (M3) and its comparison with other recent existing methods in contrast and visibility recovery. These measures can be formulated as follows

**e:** Let $n_0$ and $n_r$ be the cardinal numbers of the set of visible edges of the original $I_0$ image and restored image $I_r$. Then assuming **e** be the rate of the restored edges in $I_r$ that can be computed by

$$\mathbf{e} = \frac{n_r - n_0}{n_0}. \quad (27)$$

The value of **e** measures the amount of edges that are capable of being seen in $I_r$ and are concealed in $I_0$ due to haze and fog.

**r̄**: Let **r̄** be the geometric mean of the ratio of the gradient in the recovered image and in the original $r_i$ image relative to the visible edges for each pixel $i$. The **r̄** is given by

$$\bar{\mathbf{r}} = \exp\left[\frac{1}{n_r}\sum_i \log(r_i)\right]. \tag{28}$$

The value of **r̄** display the essential and distinguishing attributes of the proposed model in the recovery of image contrast and forecast the average visibility enhancement of the restoration algorithm.

$\sigma$: The measure $\sigma$ is obtained by normalizing the number $n_s$ of entirely black or white pixels of the algorithm after contrast restoration on the size of the image and can be formulated as

$$\sigma = \frac{n_s}{(dim_x \times dim_y)}\ , \tag{29}$$

where the width and height of the image is $dim_x$ and $dim_y$.

**Entropy:** Entropy is the corresponding states of intensity level adapted by the individual pixels in an image and is a measurement of image information content which make sense of the average precariousness of a collection of facts. It is used in the quantitative analysis, evaluation and providing better comparison of the image details. The higher entropy value result indicates more image details information.

In other words, a high entropy value means that the average amount of information in the image is high, which indicates there is a lot of information about the features or corners. Hence, the higher the entropy value of the image, the finer the contrast. The 2-D information entropy of the image reflects the average amount of information in the image, its value range is [0 8]. The larger the value is, the more uniform the gray value of the image is.

## 5.1 Outdoor and indoor scenes

This section examines the image de-hazing performance of the proposed model (M3) with five natural scenes (outdoor and indoor) corrupted with haze or fog shown in Fig 2 ((a-forest image), (f-home outdoor image), (k-dense forest image), (p-indoor room image) and (u-indoor building image)) respectively. The first column shows the given hazy or foggy images and the rest of the columns from left to right show the corresponding visible edge maps, transmission maps, air-light maps and final de-hazing results of the proposed model (M3). The image restoration results shown in Fig 2(e), 2(j), 2(o), 2(t) and 2(y) looks natural while the details and features of the images are preserved quite well. The claim can be further confirmed by the visual inspection of the de-hazing outputs displayed in the last column of Fig 2, corresponding visible edge maps given in the 2nd column of Fig 2 and quantitative results of the four measures mentioned in Table 1. It could be found that our approach can efficiently remove haze or fog and can restore high-quality haze-free images while generating no specious edges, halos, or artifacts. Note that the transmission map $T(x) = e^{-\beta d(x)}$ reflecting by the density of haze or fog as given in the 3rd column of Fig 2 depends on the distance $d(x)$ from scene to the sensor and atmospherical degradation coefficient $\beta$ caused by the scattering of light due to haze or fog in the atmosphere and is reputed as a grading variant of the depth map in case of both homogeneous and inhomogeneous haze or fog.

## 5.2 Synthetic road scene images under heterogeneous fog

This section discusses the restore image performance of the proposed model (M3), shown in Table 2 and Fig 3(a), 3(f), 3(k) and 3(p) on several synthetic road scene images under

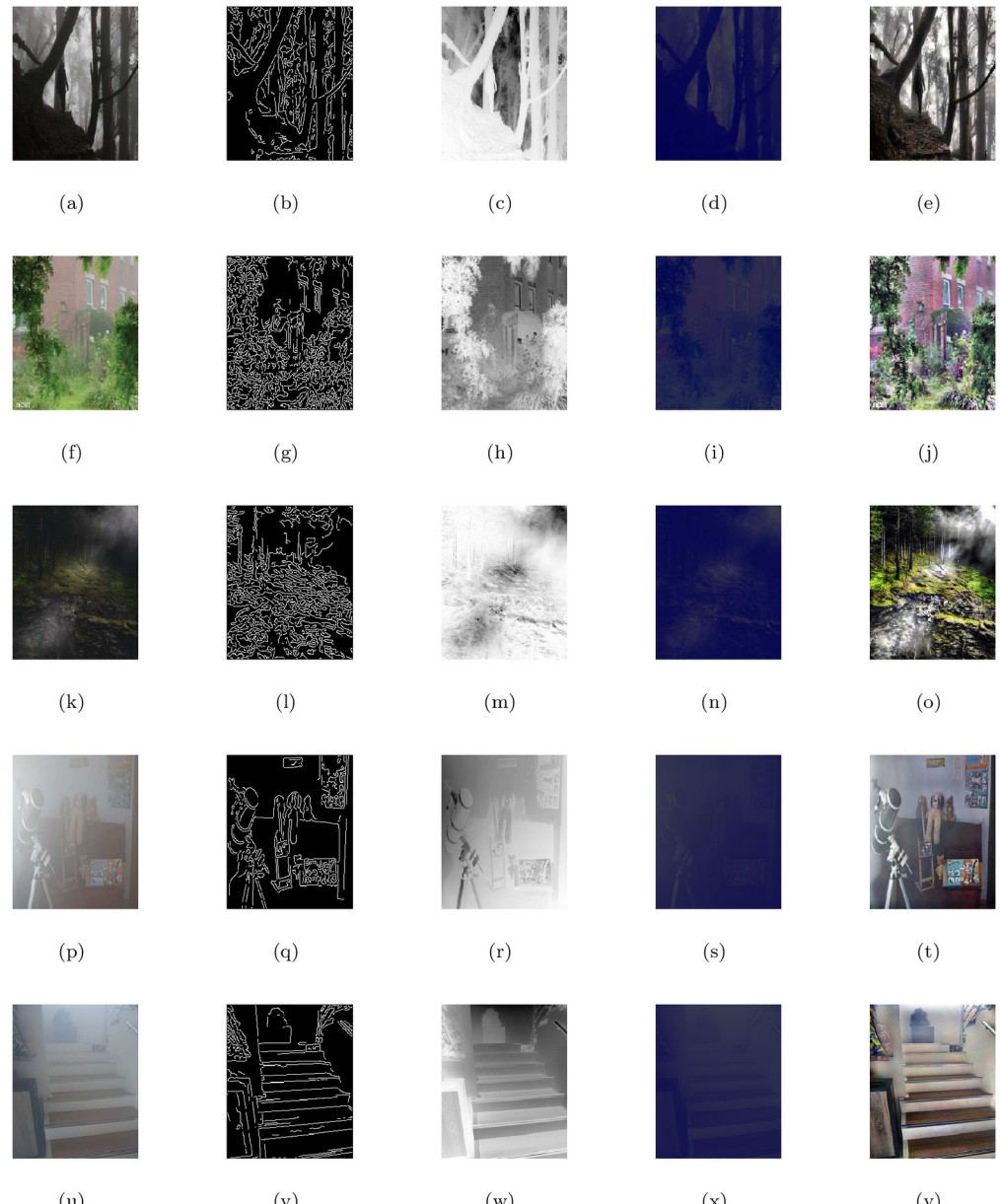

**Fig 2. De-hazing outputs of the proposed model (M3).** From left (L) to right (R), the original images (a,f,k,p,u), the corresponding visible edge maps (b,g,l,q,v), transmission maps (c,h,m,r,w), air-light maps (d,i,n,s,x) and final de-hazing results (e,j,o,t,y).

**Table 1. Quantitative results of the model-M3 under e, $\bar{r}$, $\sigma$, entropy measure and CPU time for the de-hazed images of outdoor and indoor scenes.**

| Figs | e | $\bar{r}$ | $\sigma$ | Entropy | CPU time in seconds |
|---|---|---|---|---|---|
| Fig 2(e) | 0.1936 | 1.0012 | 0.0853 | 7.7198 | 28.0625 |
| Fig 2(j) | 0.1931 | 1.0010 | 0.1438 | 7.8872 | 24.9375 |
| Fig 2(o) | 0.1363 | 1.0008 | 0.1403 | 7.6878 | 21.0156 |
| Fig 2(t) | 0.0202 | 1.0011 | 0.0867 | 7.9098 | 27.6875 |
| Fig 2(y) | 0.1599 | 1.0015 | 0.0783 | 7.9119 | 25.3437 |

**Table 2. Quantitative results of the model-M3 under e, r̄, σ, entropy measure and CPU time for the de-hazed images of synthetic road scene images under heterogeneous fog.**

| Figs | e | r̄ | σ | Entropy | CPU time in seconds |
|---|---|---|---|---|---|
| Fig 3(e) | 0.6646 | 1.0014 | 0.0557 | 7.9323 | 29.6751 |
| Fig 3(j) | 0.1777 | 1.0011 | 0.1087 | 7.9047 | 28.3568 |
| Fig 3(o) | 0.0930 | 1.0008 | 0.0925 | 7.8864 | 22.2837 |
| Fig 3(t) | 0.2359 | 1.0012 | 0.0869 | 7.8488 | 20.3845 |

heterogeneous fog respectively. In the Fig 3, the first column displays the given road scene foggy images and the rest of the columns from left to right show the associated visible edge maps ((b), (g),(l),(q)), transmission maps ((c),(h),(m),(r)), air-light maps ((d),(i),(n),(s)) and restoration results ((e),(j),(o),(t)) of the proposed model (M3). The proposed de-hazing method (M3) can restore highly improved images while maintaining very good image details and producing no artifacts, as can be seen from the result provided in Fig 3. Furthermore, the restoring results of

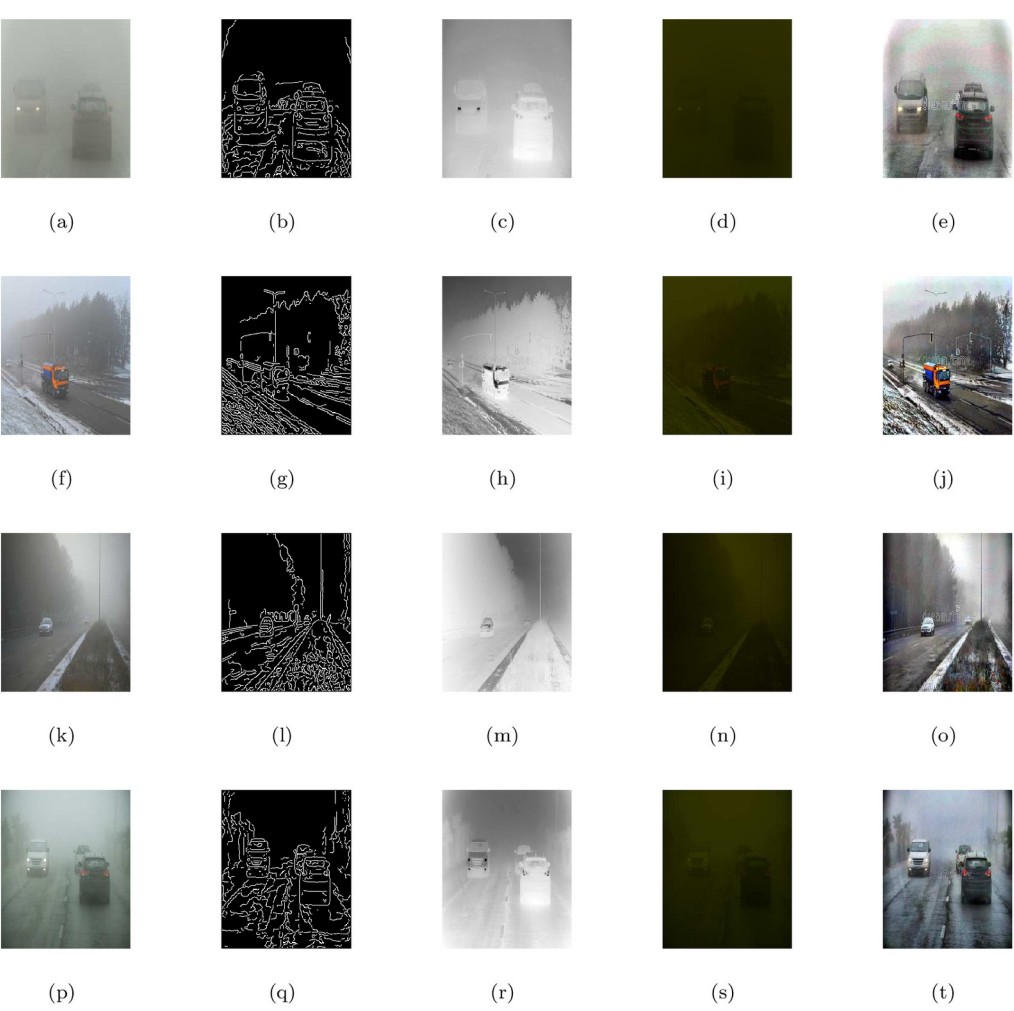

(a) (b) (c) (d) (e)

(f) (g) (h) (i) (j)

(k) (l) (m) (n) (o)

(p) (q) (r) (s) (t)

**Fig 3. De-hazing results of the proposed model (M3).** From L to R, the original images (a,f,k,p), the corresponding visible edge maps (b,g,l,q), transmission maps (c,h,m,r), air-light maps (d,i,n,s) and final de-hazing results (e,j,o,t).

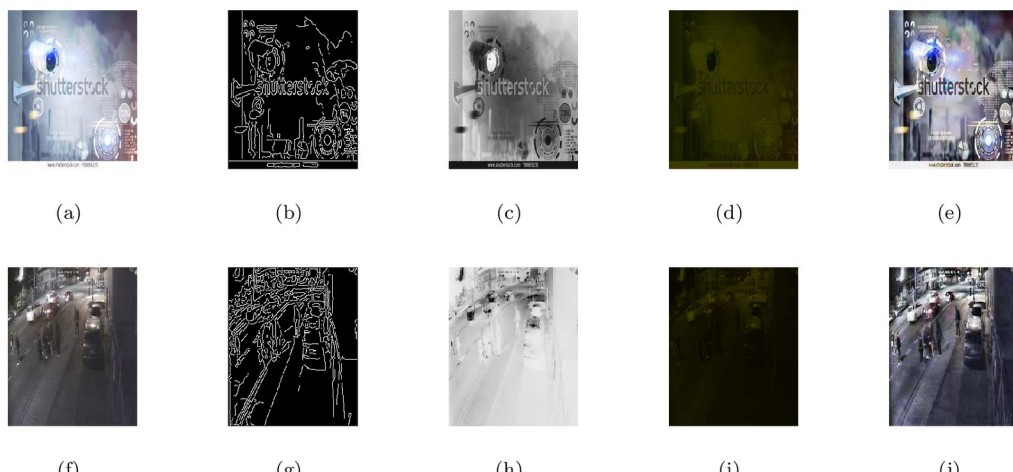

**Fig 4. De-hazing results of the proposed model (M3).** From L to R, the original images (a,f), the corresponding visible edge maps (b,g), transmission maps (c,h), air-light maps (d,i) and final de-hazing results (e,j).

the proposed model in Fig 3 and quantitative results of the four measures given in Table 2 show that the results of de-hazing can be efficiently used to treat the inland river image.

## 5.3 Realistic CCTV camera captured images under heterogeneous haze

In this section, the image visibility restoration performance of the proposed method (M3) was tested for images from CCTV cameras recorded under hazy weather conditions. In the Fig 4, the first column shows the given CCTV based hazy scenes and the remaining of the columns from left to right show the associated visible edge maps ((b),(g)), transmission maps ((c),(h)), air-light maps ((d),(i)) and restoration results ((e),(j)) of the proposed model (M3) respectively. The haze-removal outputs obtained through (M3) were checked through visual inspection shown in Fig 4 ((e),(j)) appeared brighter and more clear while keeping more visible edges and other details. Moreover, the qualitative de-hazing results of (M3) in Fig 4 and quantitative outputs in Table 3 show that the restoration results can be applied efficiently to defense and surveillance image processing.

## 5.4 Comparison with other image restoration methods

In this part, we compare the de-hazing results of our model (M3) with Cho et al. [9] (M1) and He et al. [4] (M2) algorithms. All the methods were applied to a large number of natural and synthetic data-set of hazy, foggy, and contrast-degraded images. In Figs 5 to 16, we show the comparative restore results. As shown in the de-hazing results, we can easily see that the results of our proposed model are better than other methods while maintaining edges and other image features quite well. Furthermore, we can observe also that the recovered outputs of Cho

**Table 3. Quantitative results of the model-M3 under e, r̄, σ, entropy measure and CPU time for the de-hazed images of realistic CCTV camera captured images under heterogeneous haze.**

| Figs | e | r̄ | σ | Entropy | CPU time in seconds |
|---|---|---|---|---|---|
| Fig 4(e) | 0.1152 | 1.0011 | 0.1069 | 7.8158 | 20.0007 |
| Fig 4(j) | 0.1906 | 1.0011 | 0.0849 | 7.5911 | 23.3333 |

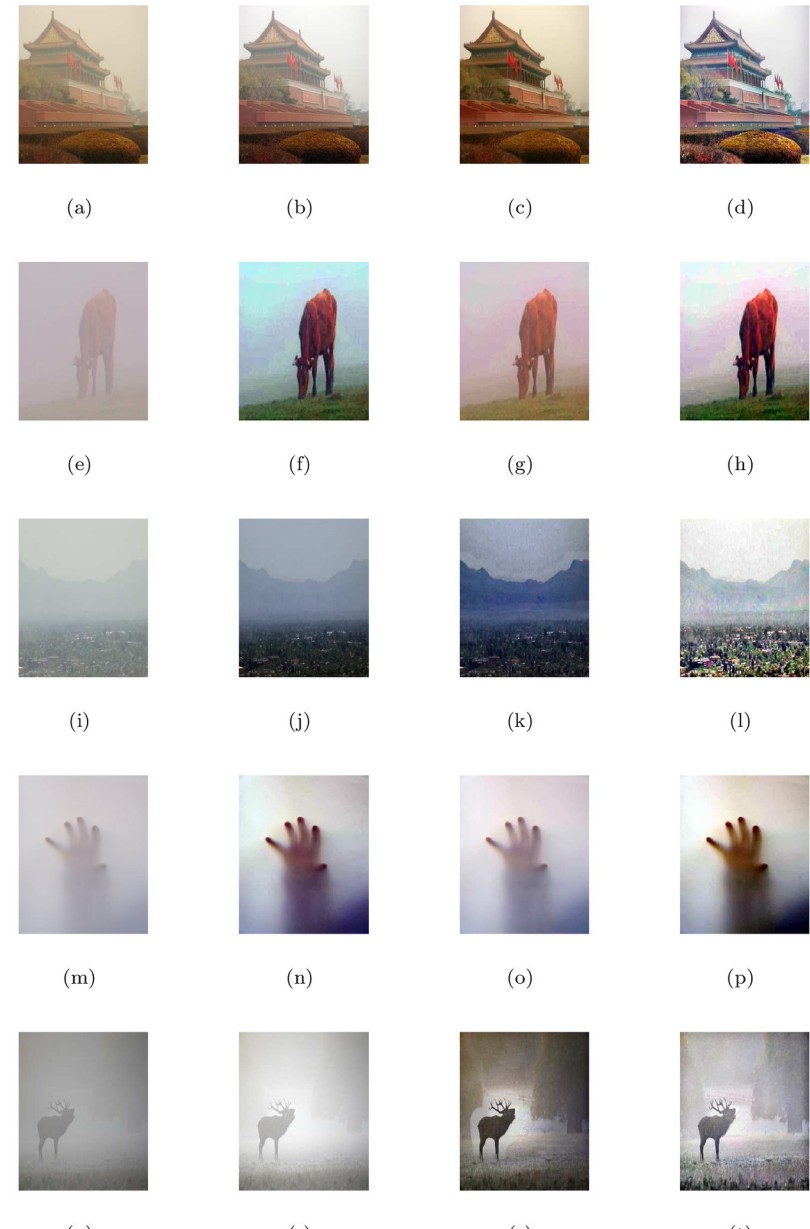

**Fig 5. De-hazing performance of the proposed model (M3) in comparison with M1 and M2.** From L to R columns represent the given degraded images, M1, M2 and M3 results, respectively.

et al. [9] and He et al. [4] approaches can increase details and visuality of the image but the color and brightness of these approaches are trivial and fiddling. The de-hazing results of Fig 5 (e) and 5(m) are just similar to Cho et al. [9] approach and the proposed approach but there is a problem of generating stair-case effects in Fig 5(e) results generated by all the three approaches. Although Cho et al. [9] approach is free of generating noise and stair-cases like artifacts in all the Figs except Fig 5(e), however, it gives very poor de-hazing results as compared to He et al. [4] and our method (M3).

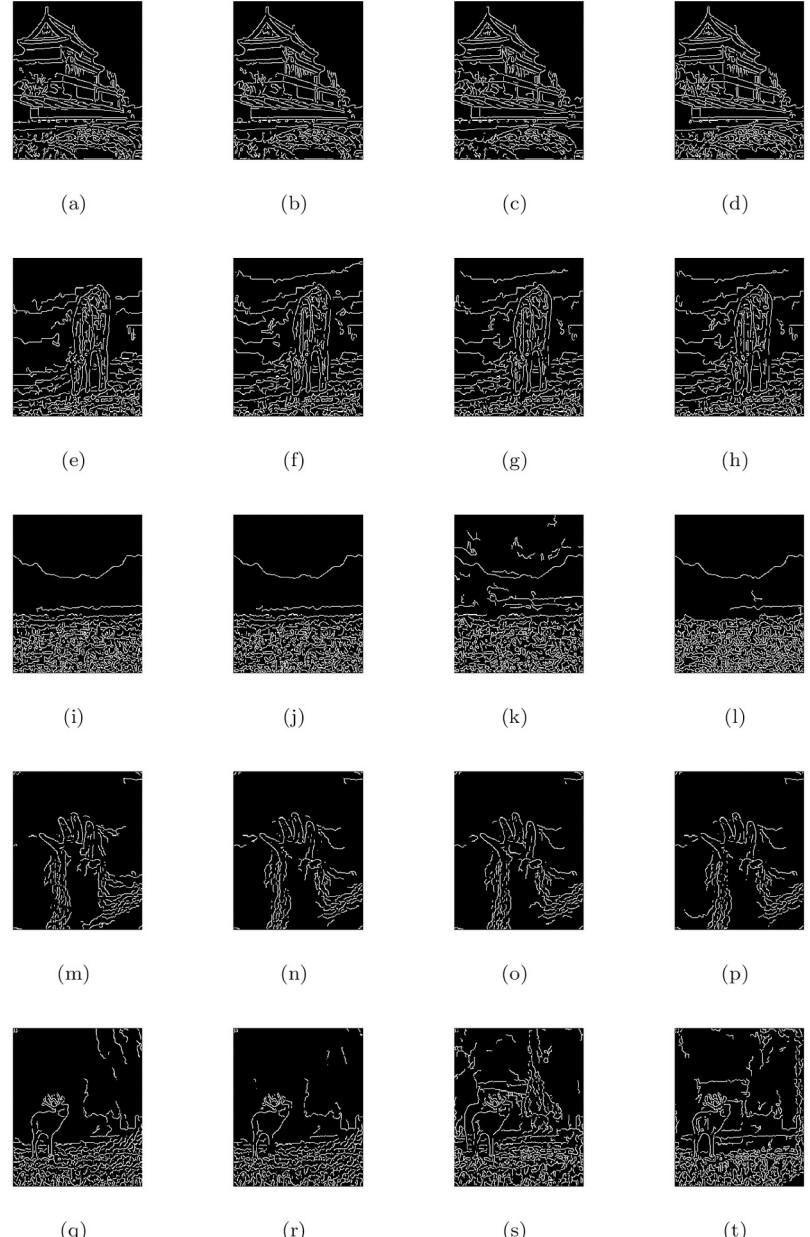

**Fig 6. Corresponding visible edge maps of Fig 5.**

In He et al. [4] de-hazing results, Figs 5(s), 7(c), 7(k), 9(s), 11(k), 13(c) and 13(o) shows some halo artifacts around the objects in the images, Figs 5(g), 5(k), 9(c) and 9(k), 13(g), 13(o) and 13(s) are less clearly visible and fade the color of the images, Fig 5(k) produces noise in large number, Fig 7(c) produces stair cases like artifacts or blocky effects and Fig 7(s) contains some blocks in the area behind the object. Due to these roughness and drawback, He et al. [4] approach generates artificial and spurious edges as shown in the corresponding visible edge maps of these images. Also their approach is based on the dark channel prior, it works well for the images with non-sky regions like Figs 5(c), 5(o), Fig 11(o) and especially in Fig 7(g). It may

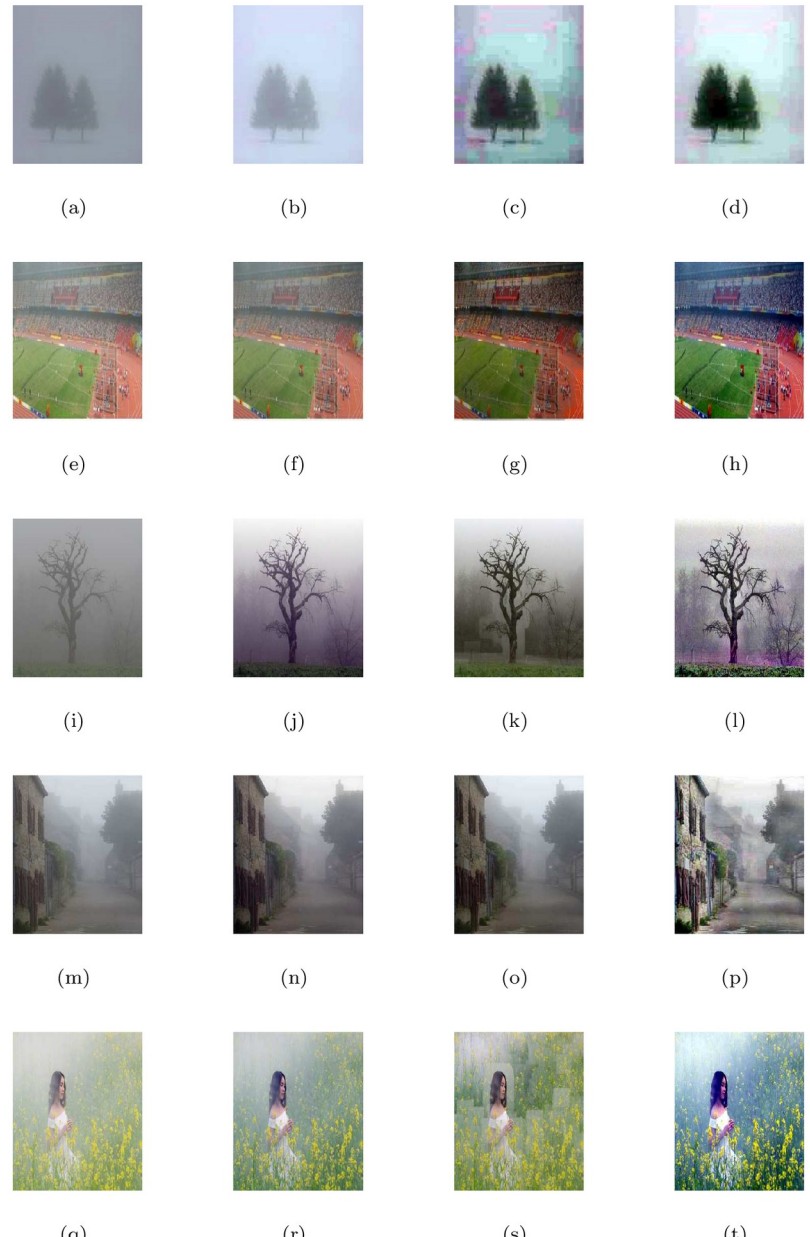

**Fig 7. De-hazing performance of the proposed model (M3) in comparison with M1 and M2 models.** From L to R columns represent the given degraded images, M1, M2 and M3 results, respectively.

fails for images containing sky regions and water (adapting shape of the sky) as shown in Fig 11((c), 11(g), 11(k) and 11(o)). Visual inspection shows that we perform more effectively with images taken at night, as shown in Fig 15 and is free of generating noise, stair-cases like artifacts and spurious edges as well. The discussion above shows how our proposed model can effectively reduce unwanted halos and our method can preserve more essential image features than other methods and can outstrip other state-of-the-art approaches in the field of image de-hazing as well.

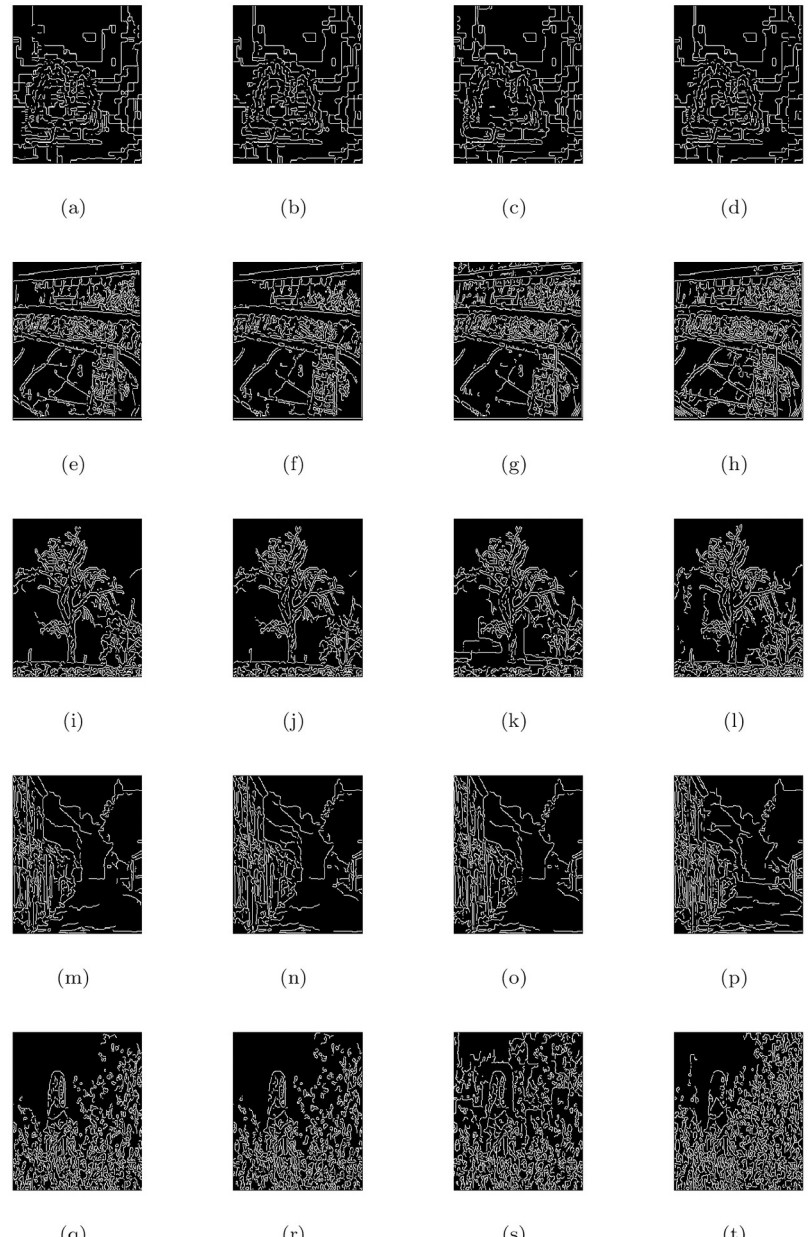

**Fig 8. Corresponding visible edge maps of Fig 7.**

## 5.5 Multiple real-world foggy image data-set (MRFID)

We compare our model (M3) de-hazing results with Cho et al. [9] model (M1), He et al. [4] model (M2) and clear images. Various experiments are performed on various images of one of the latest data-sets, multiple real-world foggy image data-set (MRFID). Some of the test results are displayed in Fig 17. The columns from left to right represent given images, M1 de-hazing results, M2 de-hazing results, proposed de-hazing results and clear images. The given images and clear images are taken from MRFID which are captured in different weather conditions. MRFID is available at http://www.vistalab.ac.cn/MRFID-for-defogging/. By visual inspection

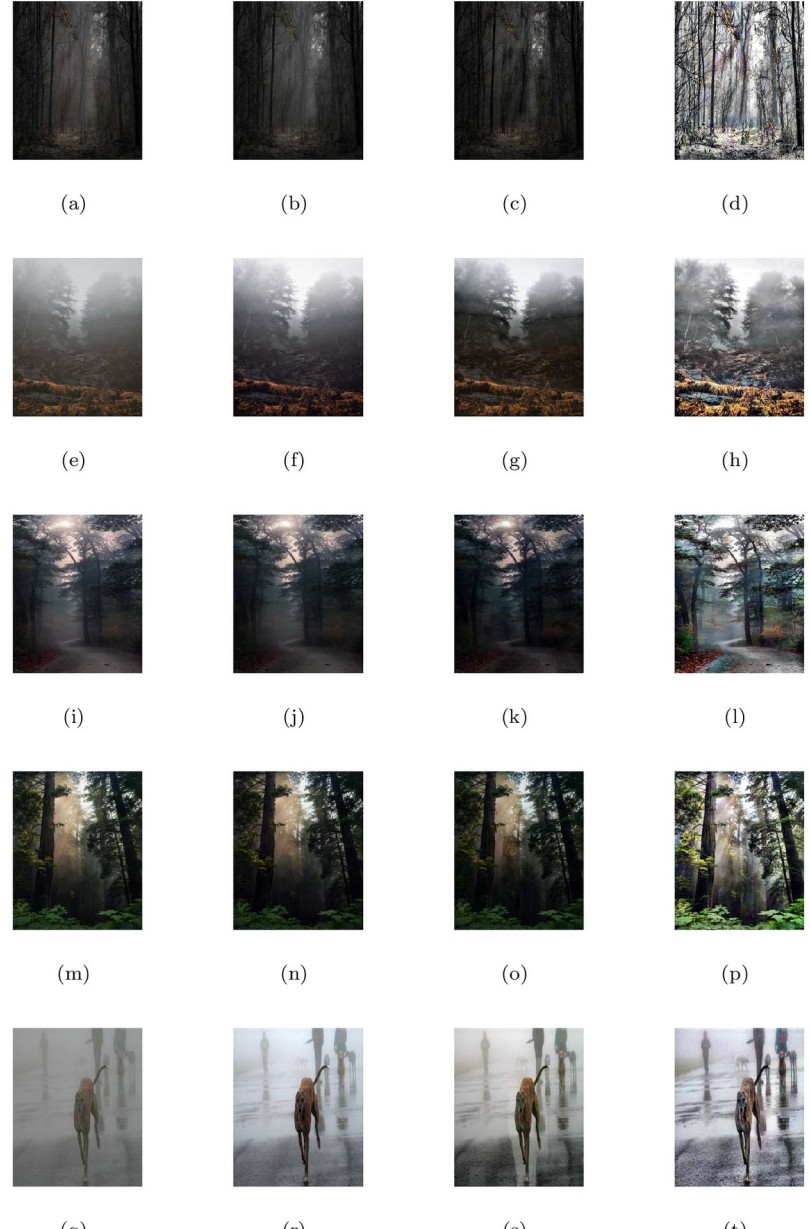

**Fig 9. Restoration performance of the proposed model (M3) in comparison with M1 and M2.** From L to R columns represent the given images, Cho et al. [9] (M1) results, He et al. [4] (M2) results and proposed model results, respectively.

of the experimental results on MRFID images, one can analyze that the de-hazing results of the M3-model is much better than M1 and M2 models and have a little difference with clear images.

## 5.6 Quantitative assessment

In this part, we quantify, analyze and compare the results of a proposed model (M3) of the de-hazing results with Cho et al. [9] (M1) and He et al. [4] (M2) methods given in Table 4 by

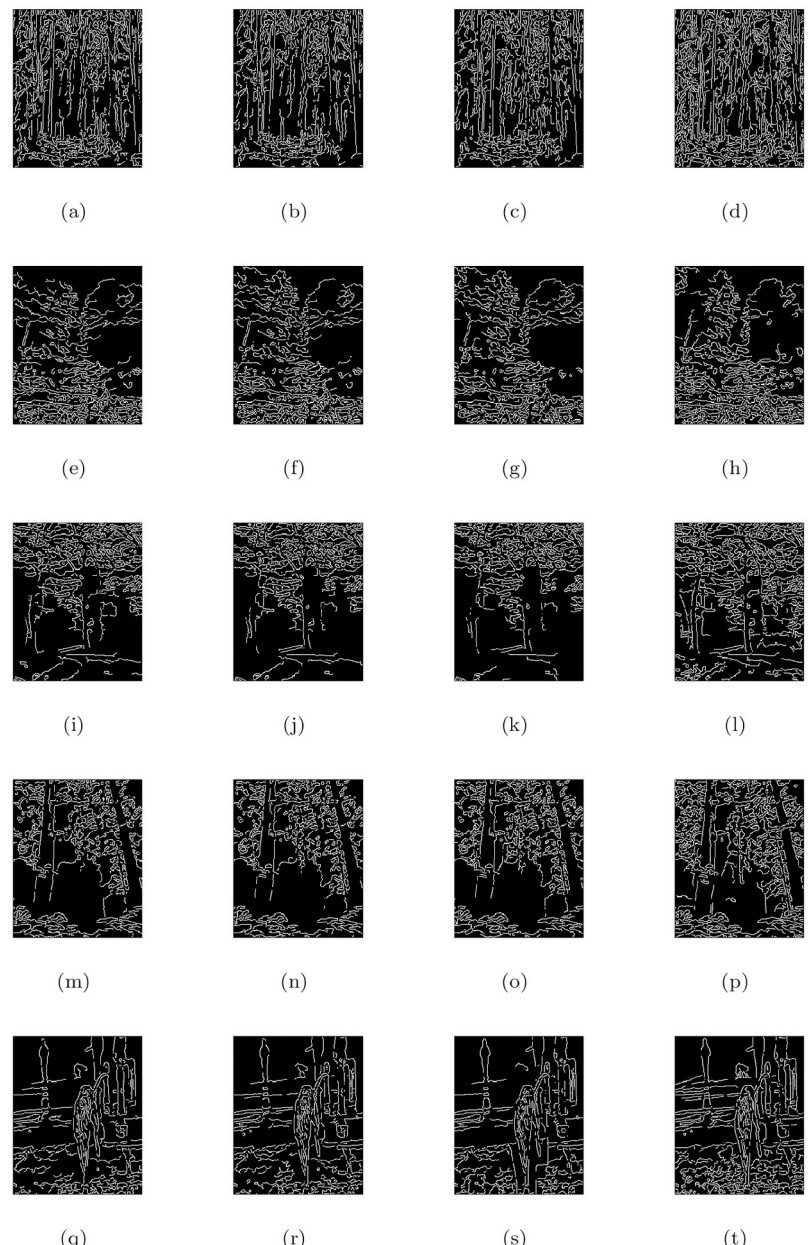

**Fig 10. Corresponding visible edge maps of Fig 9.**

showing the mean and standard deviation of the indicator values ($e$, $\bar{r}$, $\sigma$) and entropy post de-hazed values of the scenes tested in Figs 5 to 16 respectively. The indicator values ($e$, $\bar{r}$, $\sigma$), and entropy show the amount of corresponding newly visible edges, contrast and visibility enhancement, completely black or white pixels of the algorithms and states of intensity level adapted by the individual pixels of the restored images. By analyzing the values of indicators ($e$, $\bar{r}$, $\sigma$) and entropy post de-hazed images, we can conclude that the M3-method performs better and preserve the concealed edges of the images due to the fact of limiting contrast and visibility enhancement by removing haze and fog satisfactorily as compared to Cho et al. [9] and

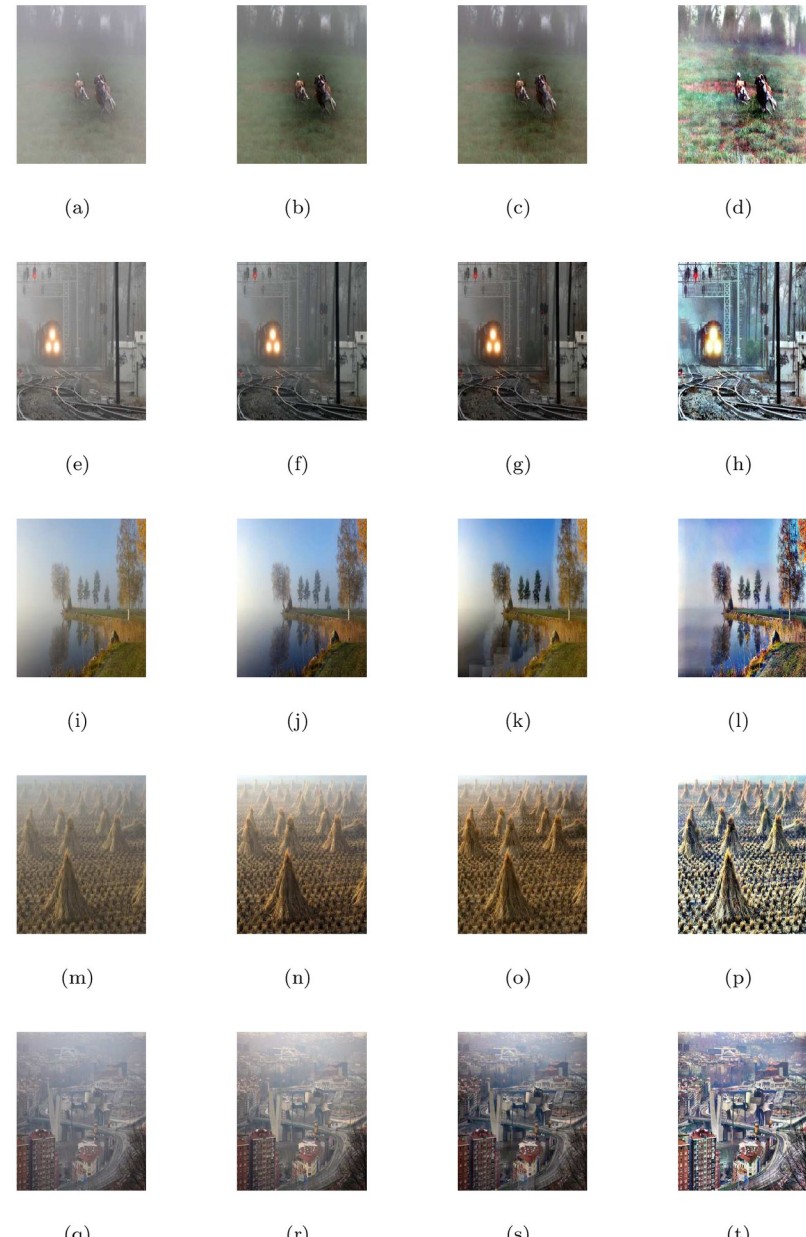

**Fig 11. De-hazing performance of the proposed model in comparison with M1 and M2.** From L to R columns represent the given images, M1, M2 and proposed model results, respectively.

He et al. [4] methods. In some of the results, He et al. [4] method yields high values of $e$ than the proposed model as in Figs (5(i), 7(i) and 7(q)) but by inspection of the corresponding visible edge maps of these Figs (6(i), 8(i) and 8(q)), it is clear that they produce spurious edges. Cho et al. [9] and He et al. [4] methods can't remove the haze and fog completely in some of the images due to which the visible edges of the images remain concealed.

Where **s.d** is the standard deviation. The proposed model has the indicator $\sigma$ values close to zero showing no existence of entirely black or white pixels comparing to state of the arts methods.

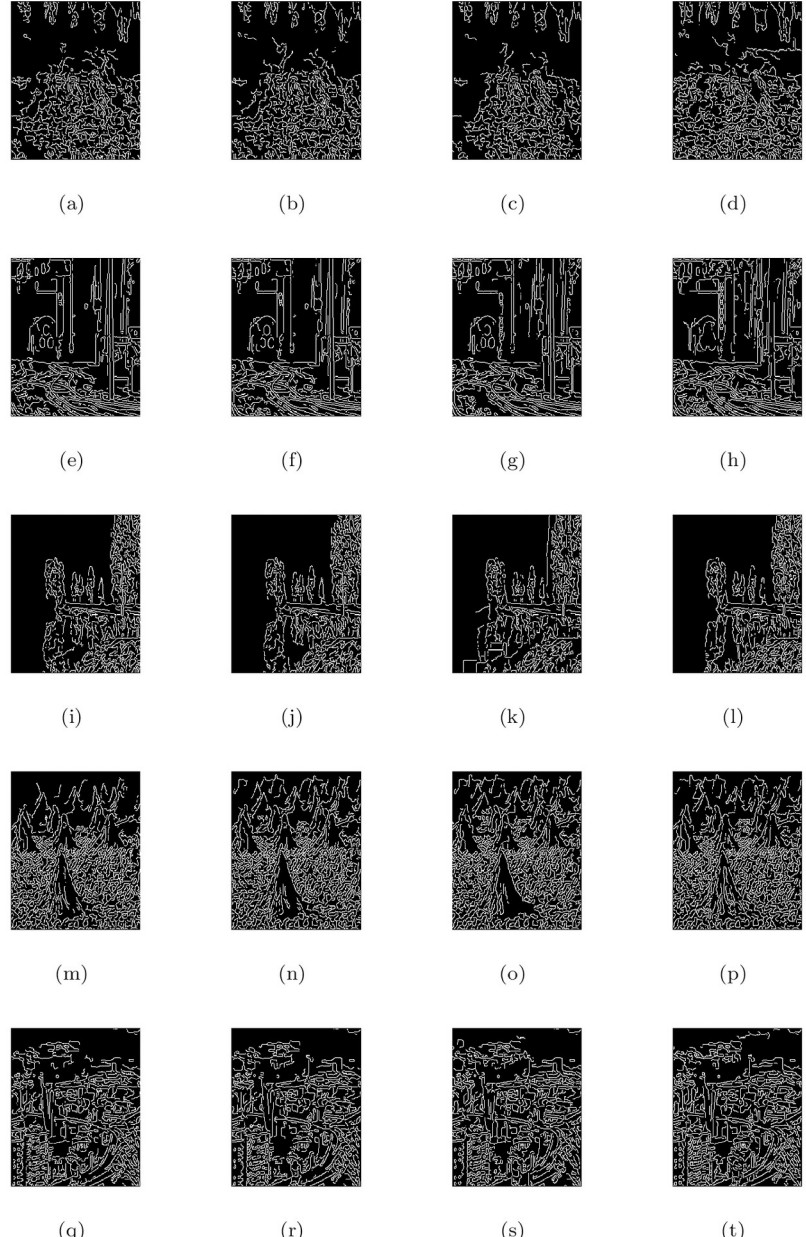

**Fig 12. Corresponding visible edge maps of Fig 11.**

## 6 Conclusion, applications and future work

This work is intended for the restoration of hazy or foggy images through the new Gaussian curvature of image surface based variation model combined with dark channel prior (DCP). The proposed method (M3) first estimates the atmospheric veil using the DCP. The transmission map is then converted to the high-quality depth map, with which the improved proposed model can be framed to obtain the haze or fog free image for both (indoor and outdoor) realistic and synthetic data set images. The proposed method can be employed for color and grey

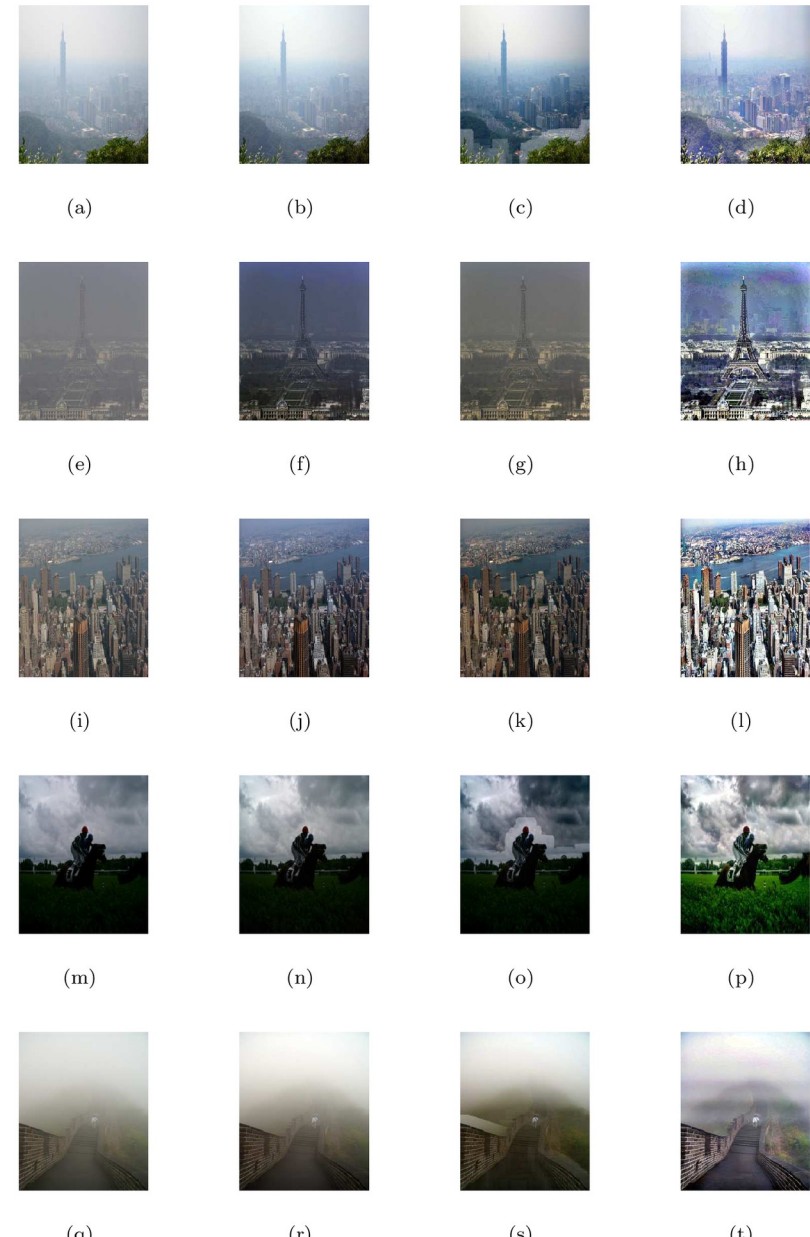

**Fig 13. De-hazing performance of the proposed model in comparison with existing models.** From L to R, columns represent the given images, Cho et al. [9] results, He et al. [4] results and proposed model results, respectively.

images. The resulting augmented Lagrangian functional was efficiently solved utilizing the augmented Lagrangian method and a special minimization procedure is adopted as well. We also employ FFT to compute numerically the system of PDEs arisen from the minimization of the energy functional. We have tested several (indoor and outdoor) hazy and foggy images in order to validate the effectiveness of the proposed method. We also compared our results with other state-of-the-art de-hazing methods. The test results have demonstrated that the

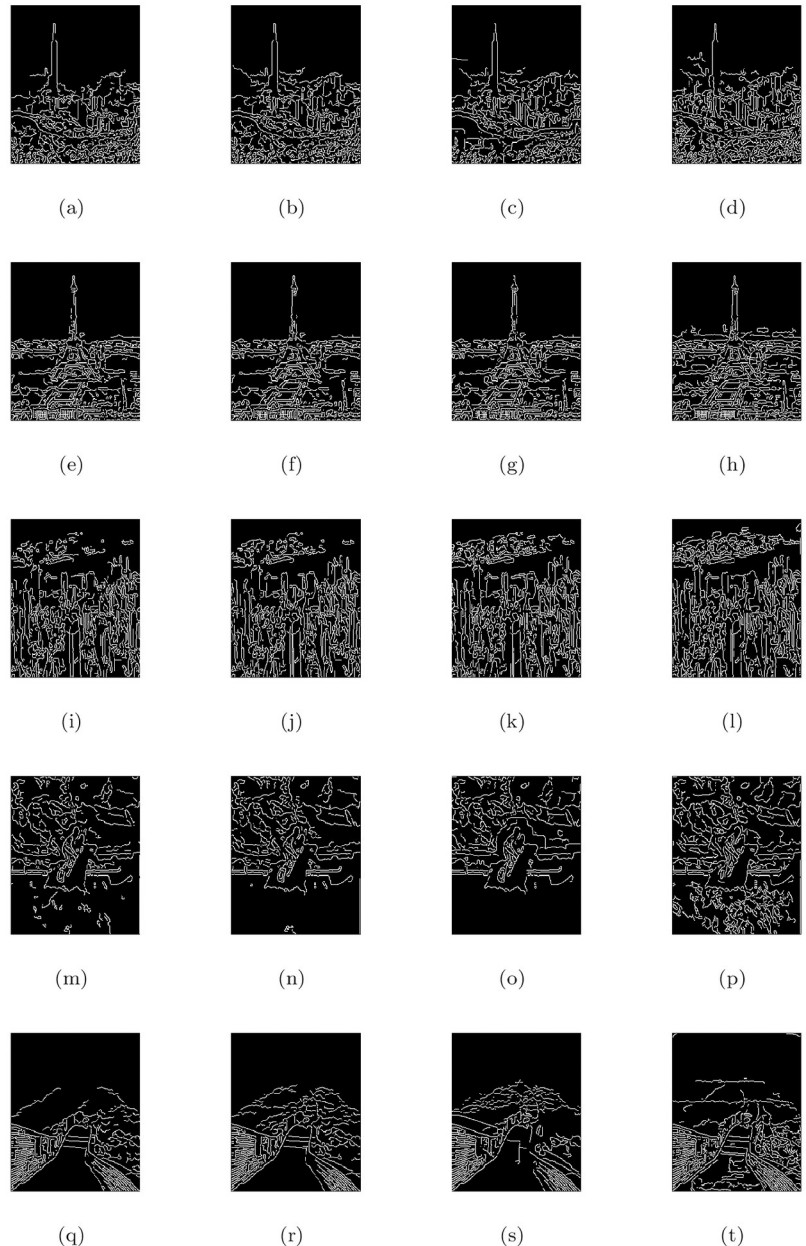

**Fig 14. Corresponding visible edge maps of Fig 13.**

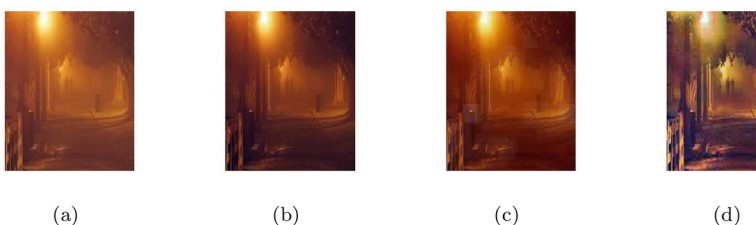

**Fig 15. De-hazing performance of the M3 de-hazing model in comparison with the existing models and clear images.** From L to R, columns represent the given images, Cho et al. [9] results, He et al. [4] results, M3-model results and clear images, respectively.

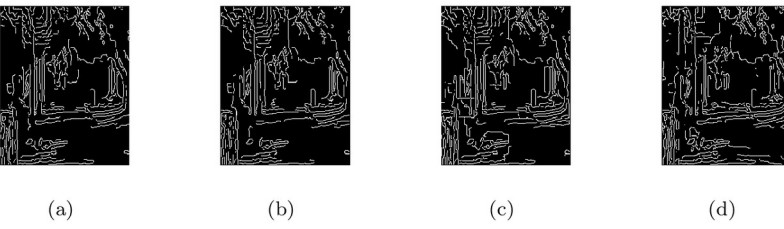

(a)    (b)    (c)    (d)

**Fig 16. Related visible edge maps of Fig 15.**

(a)    (b)    (c)    (d)    (e)

(f)    (g)    (h)    (i)    (j)

(k)    (l)    (m)    (n)    (o)

(p)    (q)    (r)    (s)    (t)

(u)    (v)    (w)    (x)    (y)

**Fig 17. De-hazing performance of the proposed de-hazing model in comparison with the existing models.** From L to R, columns represent the given images, Cho et al. [9] results, He et al. [4] results and proposed model results, respectively.

**Table 4. Quantitative outputs under coefficients (e, r̄, $\sigma$) and entropy measure for the de-hazed images of Figs 5 to 16.**

| odels | e<br>mean ± s.d | r̄<br>mean ± s.d | $\sigma$<br>mean ± s.d | Entropy<br>mean ± s.d |
|---|---|---|---|---|
| Cho et al. [9] Model (M1) | 0.0374 ± 0.0308 | 1.0014 ± 3.760 × 10⁻⁴ | 0.1963 ± 0.2561 | 7.0007 ± 0.5324 |
| He et al. [4] Model (M2) | 0.0497 ± 0.0383 | 1.0013 ± 4.940 × 10⁻⁴ | 0.1156 ± 0.0310 | 7.0054 ± 0.5141 |
| Proposed Model (M3) | 0.2436 ± 0.2021 | 1.0011 ± 2.767 × 10⁻⁴ | 0.0236 ± 0.0162 | 7.7070 ± 0.2823 |

M3-method is more effective while preserving sharp edges, contrast and other image details during the haze or fog removal process quite well than other comparing methods.

**Applications**:

The applications of the proposed method may be extended to cover image segmentation, image inpainting, inland river image processing, road scenes image processing under homogeneous and heterogeneous haze or fog, defense and surveillance images, underwater image processing, and hazy or foggy videos processing.

**Future work:**

The next work involves developing the methodology for structure detectors to indicate the structures in distinct directions and scales and then to improve the performance of the proposed method. We will also continue to deeply analyze the method proposed and demonstrate the convergence of the scheme. Some fast numerical schemes for the PDE system could be considered in future work as a result of minimizing the augmented Lagrangian functional.

## Supporting information

**S1 File.**
(ZIP)

## Author Contributions

**Conceptualization:** Muhammad Arif, Asmat Ullah, Hena Rabbani.

**Data curation:** Hadia Atta.

**Formal analysis:** Noor Badshah, Tufail Ahmad Khan, Asmat Ullah, Hena Rabbani, Hadia Atta.

**Funding acquisition:** Hena Rabbani, Hadia Atta, Nasra Begum.

**Investigation:** Muhammad Arif, Noor Badshah, Asmat Ullah, Nasra Begum.

**Methodology:** Hadia Atta, Nasra Begum.

**Project administration:** Nasra Begum.

**Software:** Asmat Ullah.

**Supervision:** Tufail Ahmad Khan.

**Validation:** Noor Badshah.

**Writing – review & editing:** Asmat Ullah.

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
