## [Decision Letter · Decision Letter 0]

18 Aug 2022

PONE-D-22-17575A new Gaussian curvature of the image surface based variational model for haze or fog removalPLOS ONE

Dear Dr. Ullah,

Thank you for submitting your manuscript to PLOS ONE. After careful consideration, we feel that it has merit but does not fully meet PLOS ONE’s publication criteria as it currently stands. Therefore, we invite you to submit a revised version of the manuscript that addresses the points raised during the review process.

Based on the reviewers' comments, the main concern is about the comparative study and Matlab experiments. Please consider all the comments from reviewers and re-submit the revised manuscript after a major revision. Basically, a proof reading would be helpful to enhance the readability of the paper.

We look forward to receiving your revised manuscript.

Kind regards,

Qichun Zhang, PhD

Academic Editor

PLOS ONE

Journal Requirements:

"details of the funding organization and funders will be provided later."

4. We note that Figure 7 includes an image of a participant in the study.

5. Please ensure that you refer to Figure 9, 10, 12, 13 and 14 in your text as, if accepted, production will need this reference to link the reader to the figure.

Additional Editor Comments:

Based on the reviewers' comments, the main concern is about the comparative study and Matlab experiments. Please consider all the comments from reviewers and re-submit the revised manuscript after a major revision. Basically, a proof reading would be helpful to enhance the readability of the paper.

Reviewers' comments:

Reviewer's Responses to Questions

**Comments to the Author**

1. Is the manuscript technically sound, and do the data support the conclusions?

Reviewer #1: Partly

Reviewer #2: Yes

2. Has the statistical analysis been performed appropriately and rigorously? 

Reviewer #1: No

Reviewer #2: Yes

3. Have the authors made all data underlying the findings in their manuscript fully available?

Reviewer #1: No

Reviewer #2: No

4. Is the manuscript presented in an intelligible fashion and written in standard English?

Reviewer #1: No

Reviewer #2: Yes

5. Review Comments to the Author

Reviewer #1: This paper investigates an interesting topic however the following comments should be further taken into account.

1) The contribution and motivation is not clearly summarised. What is the main difference of the presented M3 comparing to M1 and M2.

2) Algorithm 1 is given based on Eqs 21-26. However it is difficult to reproduce the results. Please note that the first step we need to know the value of A to determine q, while we further need the pre-specified q to obtain A. In my opinion, there are coupled to be calculated. Please give the true pseudo code to make the algorithm reproducible.

3) Please compare the performance between various models and distinguish the results from existing models.

4) Please show the model dynamical training process, for example, the cost function of entropy should be decreased during the training. The following papers would be helpful: doi.org/10.1080/21642583.2019.1588804;
doi.org/10.1016/j.neucom.2022.01.088; etc.

5) The presentation and writing need to be improved. Previous works mean the works from the same author team. If not, it should be entitled as preliminaries.

Reviewer #2: Manuscript is well-written and recent literature has been reviewed as well. The method seems to outperform some of the established models, especially produces better results in low-light/night settings and 'minimizes' halos.

My only concern is that the MATLAB code has not been made available, which would be great to review/verify the correctness of the code.

Figures are not labelled correctly, all figures miss the corresponding labels (a, b, c, ...).

6. PLOS authors have the option to publish the peer review history of their article (what does this mean?). If published, this will include your full peer review and any attached files.

Reviewer #1: No

Reviewer #2: No

---

## [Author Response · Author response to Decision Letter 0]

23 Oct 2022

Reply to reviewers comments have been submitted with a separate documents.

---

## [Decision Letter · Decision Letter 1]

21 Feb 2023

A new Gaussian curvature of the image surface based variational model for haze or fog removal

PONE-D-22-17575R1

Dear Dr. Ullah,

We’re pleased to inform you that your manuscript has been judged scientifically suitable for publication and will be formally accepted for publication once it meets all outstanding technical requirements.

Kind regards,

Kapil Kumar Nagwanshi, PhD

Academic Editor

PLOS ONE

Additional Editor Comments (optional):

Reviewers' comments:

Reviewer's Responses to Questions

**Comments to the Author**

1. If the authors have adequately addressed your comments raised in a previous round of review and you feel that this manuscript is now acceptable for publication, you may indicate that here to bypass the “Comments to the Author” section, enter your conflict of interest statement in the “Confidential to Editor” section, and submit your "Accept" recommendation.

Reviewer #2: All comments have been addressed

Reviewer #3: All comments have been addressed

2. Is the manuscript technically sound, and do the data support the conclusions?

Reviewer #2: Yes

Reviewer #3: Yes

3. Has the statistical analysis been performed appropriately and rigorously? 

Reviewer #2: Yes

Reviewer #3: (No Response)

4. Have the authors made all data underlying the findings in their manuscript fully available?

Reviewer #2: Yes

Reviewer #3: Yes

5. Is the manuscript presented in an intelligible fashion and written in standard English?

Reviewer #2: Yes

Reviewer #3: Yes

6. Review Comments to the Author

Reviewer #2: (No Response)

Reviewer #3: All comments are incorporated in Revised manuscript. Revised manuscript is now accepted for possible publication

7. PLOS authors have the option to publish the peer review history of their article (what does this mean?). If published, this will include your full peer review and any attached files.

Reviewer #2: No

Reviewer #3: No

---

## [Editor Report · Acceptance letter]

10 Mar 2023

PONE-D-22-17575R1 

A new Gaussian curvature of the image surface based variational model for haze or fog removal 

Dear Dr. Ullah:

I'm pleased to inform you that your manuscript has been deemed suitable for publication in PLOS ONE. Congratulations! Your manuscript is now with our production department. 

Kind regards, 

on behalf of

Dr. Kapil Kumar Nagwanshi 

Academic Editor

PLOS ONE